# Mitochondrial stress-induced GFRAL signaling controls diurnal food intake and anxiety-like behavior

Carla Igual Gil[1,2] , Bethany M Coull[3,7], Wenke Jonas[4,5], Rachel N Lippert[3,5,7], Susanne Klaus[1,2,*] , Mario Ost[1,6,*]

**Growth differentiation factor 15 (GDF15) is a mitochondrial stress-induced cytokine that modulates energy balance in an endocrine manner. However, the importance of its brainstem-restricted receptor GDNF family receptor alpha-like (GFRAL) to mediate endocrine GDF15 signaling to the brain upon mitochondrial dysfunction is still unknown. Using a mouse model with muscle-specific mitochondrial dysfunction, we here show that GFRAL is required for activation of systemic energy metabolism via daytime-restricted anorexia but not responsible for muscle wasting. We further find that muscle mitochondrial stress response involves a GFRAL-dependent induction of hypothalamic corticotropin-releasing hormone, without elevated corticosterone levels. Finally, we identify that GFRAL signaling governs an anxiety-like behavior in male mice with muscle mitochondrial dysfunction, with females showing a less robust GFRAL-dependent anxiety-like phenotype. Together, we here provide novel evidence of a mitochondrial stress-induced muscle–brain crosstalk via the GDF15-GFRAL axis to modulate food intake and anxiogenic behavior.**

## Introduction

Growth differentiation factor 15 (GDF15) is acknowledged as a cellular stress-induced cytokine which can be expressed and secreted by multiple tissues for local auto-/paracrine or endocrine signaling (Lockhart et al, 2020; Keipert & Ost, 2021). Among the conditions in which circulating GDF15 levels are highly elevated in humans are mitochondrial disorders (Montero et al, 2016; Dominguez-Gonzalez et al, 2020; Poulsen et al, 2020; Lehtonen et al, 2021), in which they strongly correlate with disease severity (Sharma et al, 2021). Importantly, GDF15 induction has also been confirmed in numerous studies on genetically modified mouse models of mitochondrial dysfunction, including alterations in oxidative phosphorylation (OxPhos) and coupling

efficiency (Keipert et al, 2014; Keipert et al, 2020; Ost et al, 2020), long chain fatty acid import (Pereyra et al, 2020), proteostasis (Chung et al, 2017; Choi et al, 2020, 2021; Kang et al, 2021), or mitochondrial DNA maintenance (Tyynismaa et al, 2010; Wall et al, 2015). Beyond the molecular basis of possible mitochondrial defects and the induction of a cell-autonomous integrated stress response (Suomalainen & Battersby, 2018), recent studies advanced our understanding of how local mitochondrial perturbations can affect distal tissues and promote systemic metabolic effects (Bar-Ziv et al, 2020). In mice with chronically impaired mitochondrial proteostasis, endocrine signaling of GDF15 was shown to regulate systemic energy expenditure (Chung et al, 2017; Choi et al, 2020; Kang et al, 2021). Moreover, skeletal muscle mitochondrial stress via respiratory uncoupling promotes a GDF15-dependent daytime-restricted anorectic response to control whole-body energy metabolism (Ost et al, 2020). However, along these lines, little is known about the specific downstream effects and mode of action of GDF15 in pathophysiological relevant settings of mitochondrial stress, which is crucial to develop tailored therapeutics for patients with mitochondrial disease.

The unique receptor for GDF15 that is GDNF receptor alpha-like (GFRAL) is only expressed in the hindbrain (area postrema, AP, and nucleus of the solitary tract, NTS) and signals through the tyrosine kinase co-receptor RET (Emmerson et al, 2017; Hsu et al, 2017; Mullican et al, 2017; Yang et al, 2017). Importantly, evidence from pharmacological studies using recombinant GDF15 suggests the induction of food aversion, nausea, and emesis as a result of the activation of the GDF15-GFRAL pathway (Borner et al, 2020a, 2020b; Sabatini et al, 2021). With regards to downstream targets of the GDF15-GFRAL pathway, it was shown in 2007 that GDF15 injection leads to activation of hindbrain neurons and in hypothalamic areas involved in appetite regulation such as the paraventricular nucleus (PVN) (Johnen et al, 2007). Recently, two studies confirmed the activation of corticotropin-releasing hormone (CRH) neurons in the PVN via pharmacological GDF15 treatment (Worth et al, 2020; Cimino et al, 2021). Nevertheless, little is known about the downstream molecular targets and behavioral responses linked to an endogenous activation of GFRAL signaling

[1]Department of Physiology of Energy Metabolism, German Institute of Human Nutrition Potsdam-Rehbruecke (DIfE), Nuthetal, Germany [2]Institute of Nutritional Science, University of Potsdam, Potsdam, Germany [3]Department of Neurocircuit Development and Function, German Institute of Human Nutrition, Nuthetal, Germany [4]Department of Experimental Diabetology, German Institute of Human Nutrition Potsdam-Rehbruecke (DIfE), Nuthetal, Germany [5]German Center for Diabetes Research, München-Neuherberg, Germany [6]Department of Molecular Nutritional Physiology, Friedrich Schiller University Jena, Jena, Germany [7]NeuroCure Cluster of Excellence, Charité Universitätsmedizin, Berlin, Germany

Correspondence: Mario.Ost@dife.de; klaus@dife.de
*Susanne Klaus and Mario Ost contributed equally to this work.

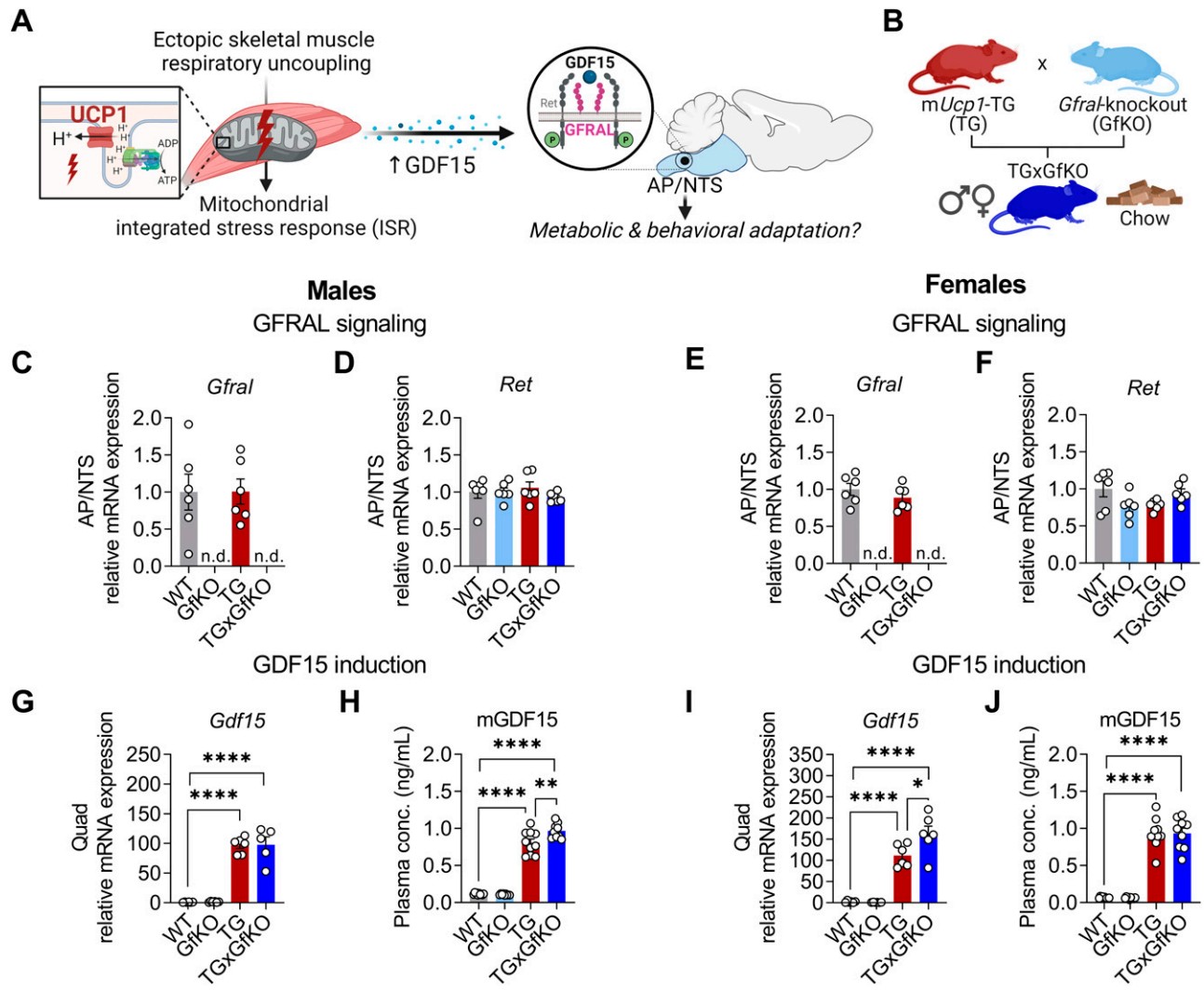

**Figure 1. Mitochondrial stress-induced GFRAL signaling induces a lean phenotype.**
**(A, B)** Schematic representations of research question and (B) experimental approach. **(C, D, E, F)** *Gfral* and (D, F) *Ret* mRNA expression in the area postrema (AP) and nucleus of the solitary tract (NTS) (n = 6). **(G, I)** Quadriceps (quad) *Gdf15* mRNA expression (n = 6). **(H, J)** Circulating GDF15 plasma levels (n = 10). Data correspond to wild-type (WT), *Gfral*-KO (GfKO), HSA-*mUcp1*-TG (TG), and HSA-*mUcp1*-TGx*Gfral*-KO (TGxGfKO) mice. The left and right panels of the figure correspond to male and female mice, respectively. Data are presented as mean + SEM. *P < 0.05; **P < 0.01; ***P < 0.001; ****P < 0.0001. Statistical test: one-way ANOVA.

pathway by mitochondrial stress. Here, using a mitochondrial dysfunction mouse model (HSA-*mUcp1*-transgenic [TG] mice) (Klaus et al, 2005; Keipert et al, 2010) with chronically elevated muscle-derived GDF15 (Ost et al, 2020), we aimed to elucidate the biological role and physiological relevance of the GFRAL receptor activation under skeletal muscle mitochondrial stress in a sex-specific manner.

## Results

### Genetic ablation of GFRAL in mice with muscle mitochondrial stress abrogates the lean phenotype but not muscle wasting

To understand the role of the receptor GFRAL in the potential metabolic and behavioral adaptation under muscle mitochondrial

stress induced by ectopic uncoupling protein 1 (UCP1) expression (Fig 1A), we crossed TG mice with whole-body *Gfral*-knockout (GfKO) mice, obtaining *Gfral*-ablated TG mice (TGxGfKO) (Fig 1B). In line with previous studies (Emmerson et al, 2017; Hsu et al, 2017; Mullican et al, 2017), male and female chow-fed GfKO mice were phenotypically undistinguishable from their wild-type (WT) littermates. Importantly, *Gfral* expression was confirmed to be confined to the hindbrain, specifically to the AP and NTS in both WT and TG mice (Fig S1). Furthermore, GFRAL ablation, as evidenced by non-detectable *Gfral* expression in the hindbrain (Fig 1C and E), did not affect expression of the co-receptor *Ret* (Fig 1D and F), which remained unchanged among genotypes in males and females. In male TG mice, loss of GFRAL did not affect muscle *Gdf15* expression but led to slightly increased circulating GDF15 (Fig 1G and H), whereas in female TG mice, muscle *Gdf15* expression was slightly

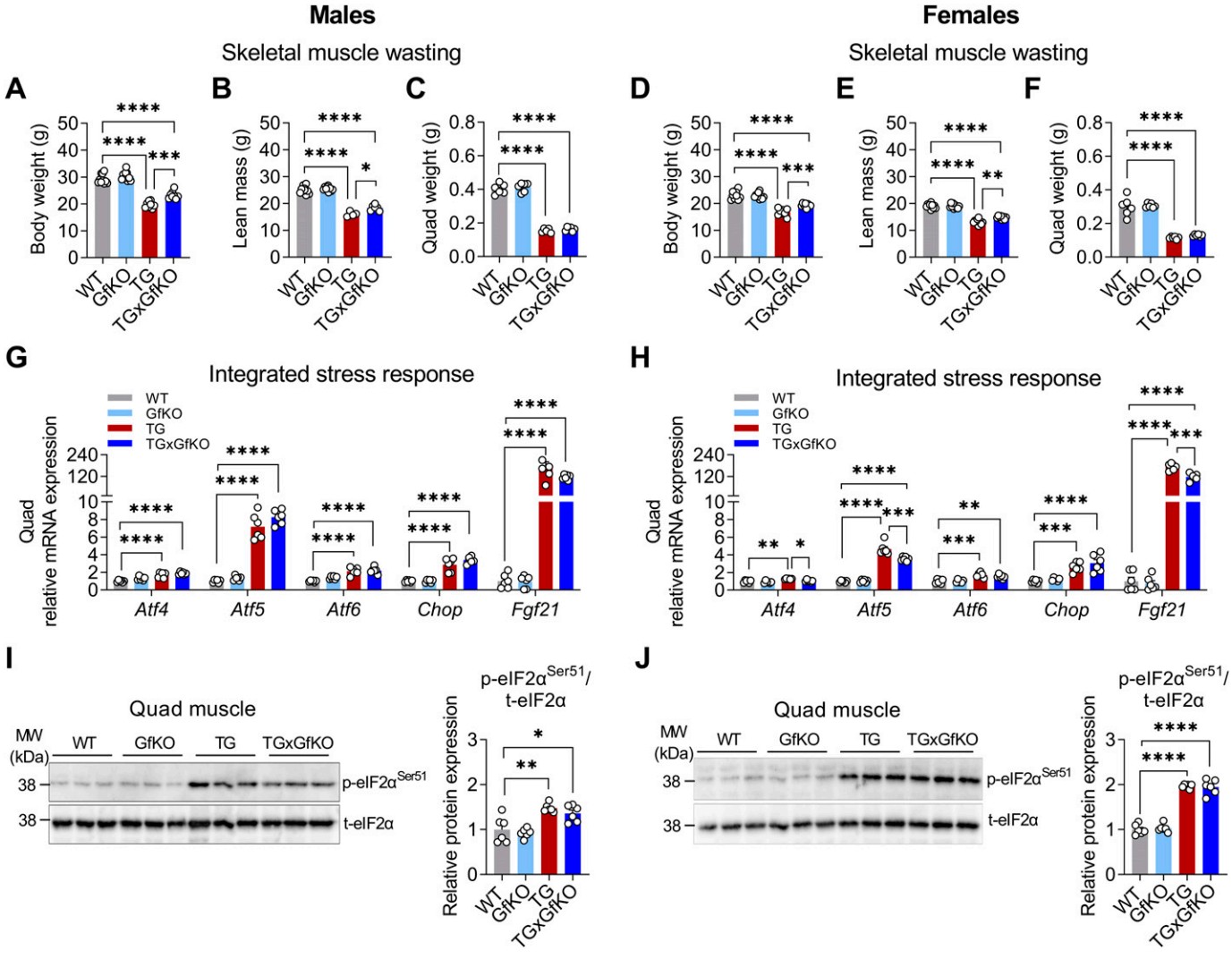

**Figure 2. Muscle wasting and mitochondrial integrated stress response are independent of GFRAL signaling.**
**(A, B, C, D, E, F)** Body weight, (B, E) body lean mass, and (C, F) quadriceps (quad) tissue weight (n = 10). **(G, H)** Quadriceps (quad) *Atf4*, *Atf5*, *Atf6*, *Chop,* and *Fgf21* gene expression (n = 6). **(I, J)** Quadriceps (quad) relative p-eIF2α$^{Ser51}$/t-eIF2α protein expression blot and quantification. Data correspond to wild-type (WT), *Gfral*-KO (GfKO), HSA-m*Ucp1*-TG (TG), and HSA-m*Ucp1*-TGx*Gfral*-KO (TGxGfKO) mice. The left and right panels of the figure correspond to male and female mice, respectively. Data are presented as mean + SEM. *$P$ < 0.05; **$P$ < 0.01; ***$P$ < 0.001; ****$P$ < 0.0001. Statistical test: one-way ANOVA.

increased by loss of GFRAL but circulating GDF15 was unaffected (Fig 1I and J).

In line with our previous data (Keipert et al, 2011; Ost et al, 2020), the body weight (BW) of male and female TG mice was reduced because of a substantially lower lean mass (LM), which was mildly restored by GFRAL ablation in male and female TGxGfKO mice (Fig 2A, B, D, and E). Importantly, mitochondrial stress-induced skeletal muscle wasting, as evidenced by largely reduced quadriceps weight, was not affected by GFRAL ablation (Fig 2C and F). Furthermore, we evaluated the involvement of GFRAL in the induction of the integrated stress response, the main suggested cellular adaptive mechanism upon mitochondrial stress. Increased muscle gene expression in TG mice of the integrated stress reponse components *Atf4*, *Atf5*, *Atf6*, and *Chop* as well *Fgf21,* a well-known endocrine mediator

involved in the mitochondrial stress response, was unaffected by the loss of GFRAL in male mice (Fig 2G). In female mice, muscle expression of *Atf4*, *Atf5*, and *Fgf21* were slightly reduced in TGxGfKO compared with TG, although still highly increased compared with WT mice (Fig 2H). Along these lines, phosphorylation of eukaryotic translation initiation factor 2α (eIF2α) was unaffected by the loss of GFRAL in both male and female mice (Fig 2I and J). Overall, these results indicate that GFRAL signaling is not involved in muscle wasting under mitochondrial stress in male and to a likely negligible extent in female mice.

Of note, loss of GFRAL restored the weight of other peripheral organs such as the liver, heart, and different fat depots (Fig S2A–F), indicating a potential role for the brainstem GFRAL activation in the control of body composition and tissue growth under chronic muscle mitochondrial stress.

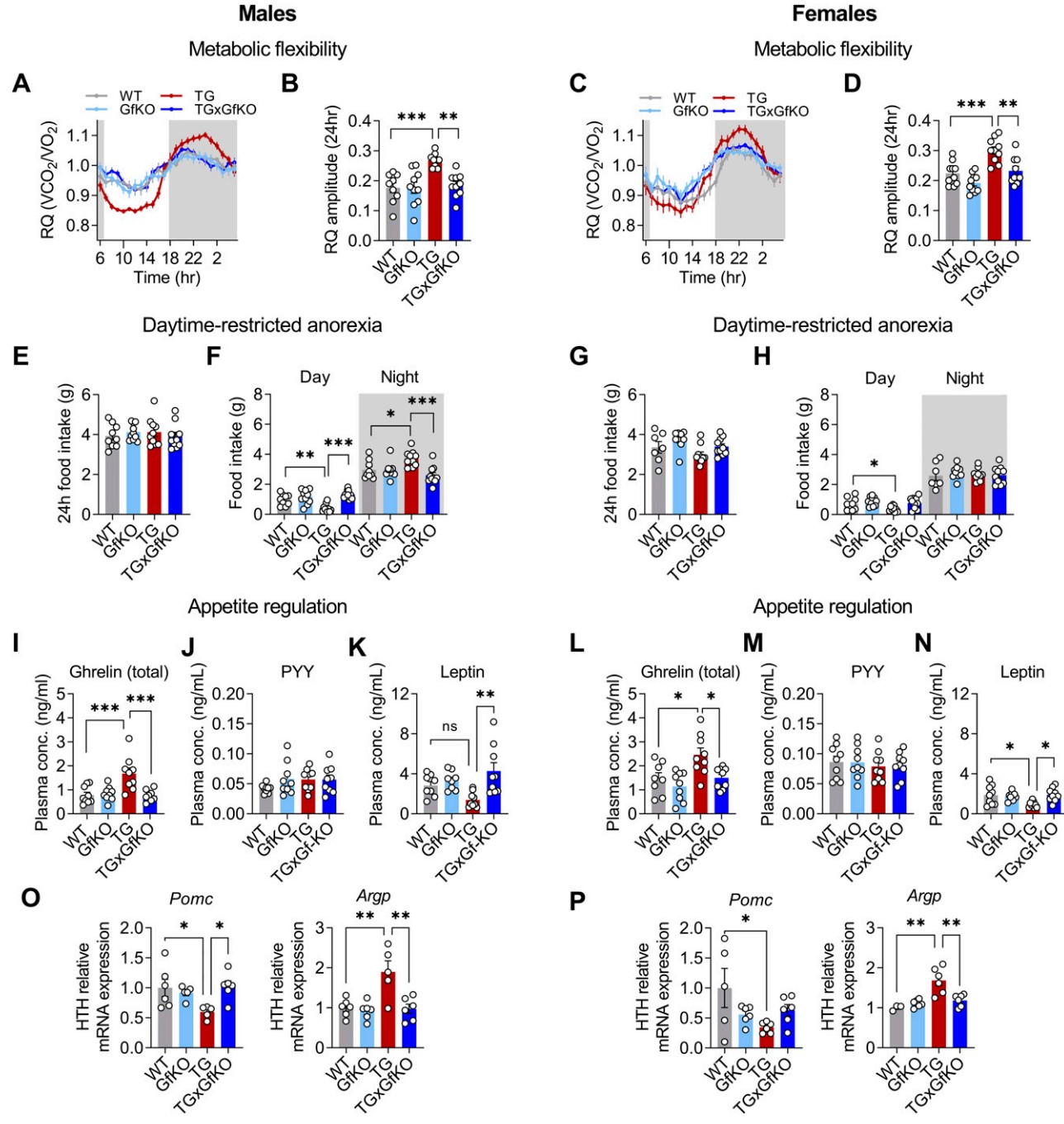

**Figure 3. GFRAL signaling induces metabolic flexibility and modulates diurnal food intake in response to mitochondrial stress.**
**(A, B, C, D)** Respiratory quotient (RQ) shown hourly over 24 h and **(B, D)** RQ amplitude quantification (n = 10). **(E, F, G, H)** Quantification of total 24 h and (F, H) day versus nighttime food intake (n = 10). **(I, J, K, L, M, N)** Circulating plasma ghrelin (total), (J, M) PYY, and (K, N) leptin at daytime (10 AM) (n = 9). **(O, P)** Hypothalamus (HTH) *Pomc* and *Agrp* gene expression (n = 6). The left and right panels of the figure correspond to male and female mice, respectively. Data correspond to wild-type (WT), *Gfral*-KO (GfKO), HSA-*mUcp1*-TG (TG), and HSA-*mUcp1*-TGx*Gfral*-KO (TGxGfKO) mice. Data are presented as mean + SEM. *$P < 0.05$; **$P < 0.01$; ***$P < 0.001$; ****$P < 0.0001$. **(B, D, E, F, G, H, I, J, K, L, M, N, O, P)** Statistical test: one-way ANOVA.

## Muscle–brain crosstalk via the GFRAL receptor mediates systemic energy metabolism and diurnal shift in feeding behavior

In a previous study, we demonstrated that muscle-derived GDF15 mediates the systemic metabolic remodeling and promotes diurnal anorexia (Ost et al, 2020), but targeted signaling from muscle to peripheral tissues remained to be elucidated. Here, we sought to investigate the involvement of the GDF15 receptor GFRAL in the systemic metabolic adaptations and daytime-restricted anorexia upon muscle mitochondrial stress. A consequence of the diurnal

variation of energy balance in TG mice is an increased systemic metabolic flexibility as evident by an increased amplitude of the respiratory quotient (RQ) (Ost et al, 2020). Here, in both male and female TG mice, increased metabolic flexibility assessed by 24-h recording and daily amplitude of the RQ was abolished by loss of GFRAL (Fig 3A–D). Thereby, TGxGfKO double-mutant mice phenocopy GDF15-ablated TG mice (Ost et al, 2020) demonstrating a muscle–brain crosstalk via the GDF15-GFRAL axis to modulate systemic energy metabolism upon mitochondrial stress. Furthermore, although total 24-h food intake remained unaffected (Fig 3E and G), GFRAL signaling proved to be completely responsible for daytime-restricted anorexia elicited by GDF15 (Fig 3F and H). Interestingly, in male but not in female TG mice, GFRAL signaling controls a nighttime increase in food intake (Fig 3F), highlighting sex-specific differences of diurnal food intake regulation under mitochondrial stress conditions. Of note, GFRAL ablation in TG mice led to a normalization of their energy expenditure phenotype (Fig S3), which appeared to be lower compared with WT when calculated in an absolute manner (Fig S3A–D) but higher when normalized to BW or LM (Fig S3E–H) in both males and females because of the reduced BW and LM in TG mice.

To further characterize molecular traits of daytime-restricted anorexia, we analyzed plasma levels of appetite regulating hormones and the gene expression pattern of the known central appetite regulators in the hypothalamus, the anorectic proopio-melanocortin (POMC), and the orexigenic agouti-related peptide (AgRP) (Gao & Horvath, 2008). We found that plasma concentrations of total ghrelin, a well-described orexigenic hormone (Inui, 2001), were increased in male and female TG mice but abrogated in TGxGfKO mice (Fig 3I and L). Interestingly, despite the reduced food intake at daytime, plasma levels of central satiety hormones peptide tyrosine–tyrosine (PYY) (Kirchner et al, 2010) and leptin (Friedman, 2019) were unaffected or lowest in male and female TG mice, respectively (Fig 3J, K, M, and N). In line with the restored weights of adipose tissue depots, plasma levels of leptin were higher in TGxGfKO versus TG mice. Finally, although hypothalamic *Pomc* expression was reduced, expression of agouti-related protein (*Agrp*) was increased in male and female TG mice, which was normalized to WT levels by GFRAL ablation (Fig 3O and P). Overall, the here observed GFRAL-dependent pattern in TG mice reflects a state of negative energy balance with increased appetite signaling aimed to increase food intake. Nevertheless, TG mice show a marked daytime-restricted anorexia (Fig 3F and H) that, together with the regulation of classical appetite/satiety modulators mentioned above, resembles an anorexia nervosa–like phenotype and suggests that the GDF15-GFRAL axis works through an alternative pathway that overrides the classic hypothalamic food intake regulation system as previously suggested (Hsu et al, 2017).

### GFRAL signaling induces hypothalamic CRH and anxiety-like behavior in response to muscle mitochondrial stress

We aimed at dissecting the underlying brain-specific downstream signaling and behavioral response upon chronic mitochondrial stress-induced GDF15-GFRAL signaling. Recent studies demonstrated the activation of CRH neurons in the PVN via pharmacological GDF15 treatment (Worth et al, 2020; Cimino et al, 2021).

Strikingly, we could show that chronic muscle mitochondrial stress promotes a consistent GFRAL-dependent increase in hypothalamic *Crh* expression in both male and female TG mice (Fig 4A and D). Interestingly, this was neither followed by an increase in pituitary gland *Pomc* expression (Fig 4B and E) nor by increased plasma corticosterone levels (Fig 4C and F), which remained unchanged in TG mice indicating no further activation of the hypothalamic–pituitary–adrenal (HPA) axis. Given the involvement of CRH in the control of the stress response and anxiety-like behavior (Reul & Holsboer, 2002), we aimed to further characterize behavioral implications of GFRAL-dependent increased hypothalamic CRH in TG mice. Performing an open field test (OFT), we observed that although total distance traveled during the OFT was not altered (Fig S4A and C), male TG mice showed a reduced number of entries in the center and time in the center that was abolished by the loss of GFRAL, whereas time in the corners remained unaffected (Fig 4G). Female TG mice, however, did not present with an increased anxiety-like phenotype during the OFT (Fig 4H). To expand on the behavioral traits of TG mice, we performed an elevated plus maze (EPM) test. Similar to the results of the OFT, male TG mice showed an increased anxiety-like behavior as evidenced by a reduced number of entries into the open arms, a decreased time in the open arms, and a decreased distance traveled in the open arms (Fig 4I) that was accompanied by a decreased distance traveled in the closed arms and an increased time freezing (Fig 4K) which was reversed in TGxGfKO mice, indicating an involvement of GFRAL signaling in inducing anxiety-like behavior in male mice. Female TG mice, however, showed a milder phenotype with only a slight decrease in the distance traveled in the open arms (Fig 4J) and an increased time freezing (Fig 4L) that was only in tendency restored in TGxGfKO mice. Along these lines, analysis of plasma corticosterone post-EPM indicated an increase in TG male mice that seemed to be, at least partially, GFRAL-dependent (Fig S4B), whereas it was not affected in female TG mice (Fig S4D), indicating an increased stress resilience of female TG mice. Altogether, these data show a weaker or negligible action of GFRAL signaling in inducing anxiety-like behavior (measured with the OFT and EPM behavioral paradigms) upon mitochondrial stress in female mice. In summary, we here demonstrate that muscle mitochondrial stress signals to the brain in a GFRAL-dependent manner to control systemic energy metabolism, as evident from increased metabolic flexibility and daytime-restricted anorexia, as well as hypothalamic signaling via CRH induction and anxiety-like behavior, at least in male mice (Fig 4M).

## Discussion

Increasing evidence supports the crucial role of mitochondria within the organism in health, during aging and disease progression (Picard & Sandi, 2021), including the view that mitochondrial stress signaling through "mitokines" is central to communicate mitochondrial dysfunction from affected tissues to peripheral target organs (Bar-Ziv et al, 2020; Klaus et al, 2021). However, crucial circulating factors, endocrine target tissues, and specific effects at

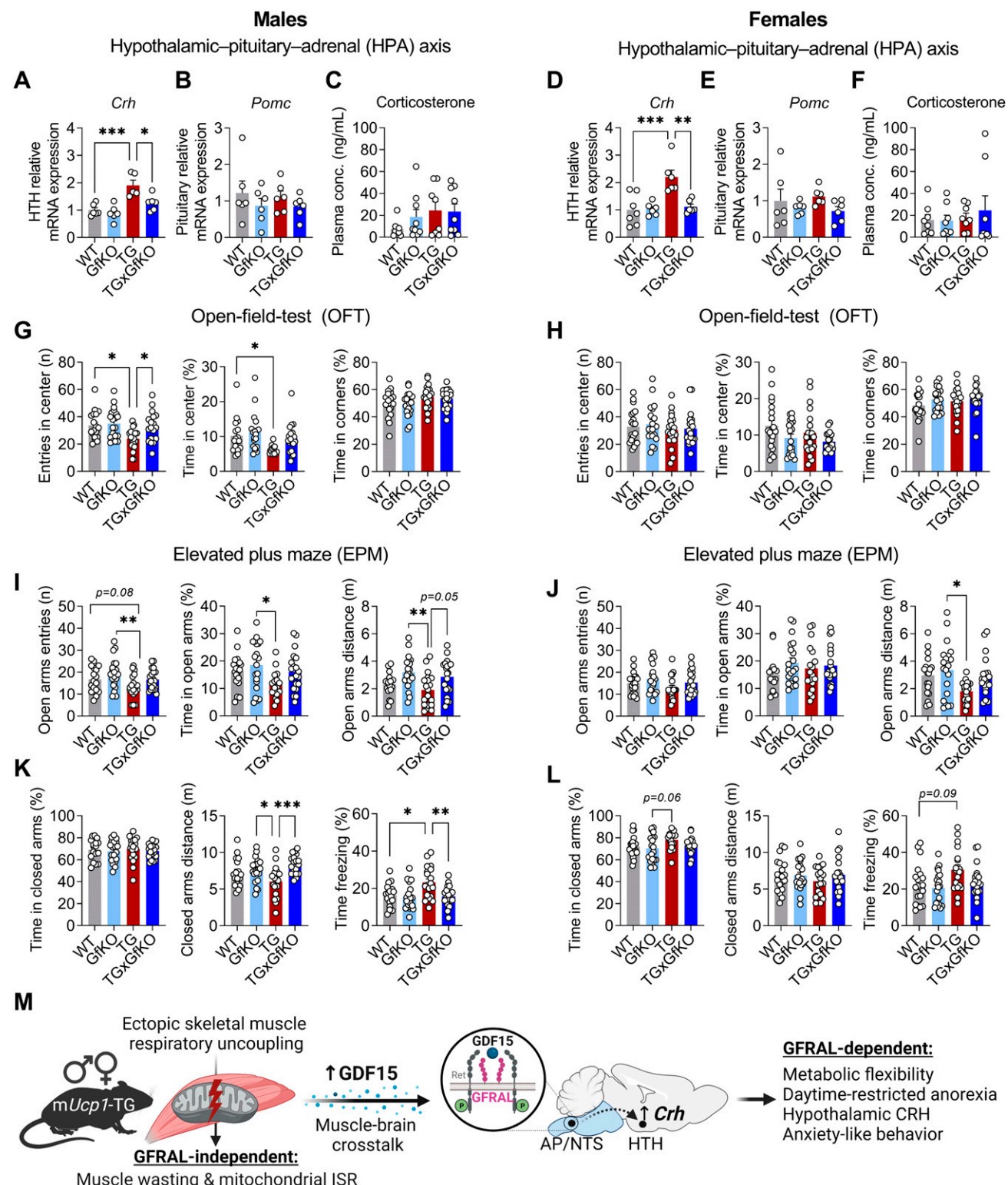

**Figure 4. Activation of GFRAL by mitochondrial stress induces hypothalamic *Crh* and anxiety-like behavior.**
**(A, D)** Hypothalamus (HTH) *Crh* expression (n = 8–9). **(B, E)** Pituitary *Pomc* expression (n = 6). **(C, F)** Plasma corticosterone levels (n = 8–9). **(G, H)** Entries in the center, time spent in the center, and time spent in the corners during an open field test (n = 16–20). **(I, J)** Open-arm entries, time spent in the open arms, and distance traveled in the open arms during an elevated plus maze test (n = 20). **(K, L)** Time spent in the closed arms, distance traveled in the closed arms, and time freezing in the closed arms during an elevated plus maze test (n = 18–20). **(M)** Schematic representation of the role of GFRAL signaling under skeletal muscle mitochondrial stress. The left and right panels

molecular, metabolic, and behavioral levels are still understood. Moreover, to the best of our knowledge, comprehensive profiling of sex-specific mitochondrial stress response pathways and endocrine signaling, particularly in female animal models, remain scarce. In mouse models of mitochondrial dysfunction, beneficial metabolic adaptation and extended lifespan were found to be more pronounced in females than males (Alfadda et al, 2004; Diaz et al, 2005). Here, we report that muscle mitochondrial stress signals to the brain via GFRAL receptor signaling to promote hypothalamic *Crh* induction in both male and female TG mice, which associates with GFRAL-dependent modulation of systemic energy metabolism, diurnal food intake, and anxiety-like behavior. However, we observed a gender bias in terms of energy balance and anxiety, with female mice showing less pronounced variations in energy balance and anxiety-like phenotype. This is in line with previous reports on gender effects in the behavior of mouse lines, showing that male mice had a higher anxiety level than their female littermates (Võikar et al, 2001), likely because of female sex hormones such as estrogens and progesterone that could affect emotions and cognition (ter Horst et al, 2012).

Next, we show that mitochondrial stress-mediated muscle wasting is induced independently of GFRAL signaling, confirming our previous data on the role of GDF15 (Ost et al, 2020). Similarly, we report here that, under mitochondrial stress conditions, loss of GFRAL leads to a partial recovery of lean and fat mass, indicating that GFRAL signaling might be instrumental in the development of a lean cachectic phenotype under conditions of mitochondrial dysfunction such as mitochondrial disease and therefore a potential target for therapeutic interventions in these patients. In line with our results, in a cancer mouse model, antibody-mediated inhibition of the GFRAL receptor proved effective in reversing cancer cachexia (Suriben et al, 2020). Furthermore, recent data show that severe myalgic encephalomyelitis/chronic fatigue syndrome (ME/CFS), a debilitating condition characterized by skeletal muscle fatigue and mitochondrial dysfunction (Morris & Maes, 2014), is associated with increased levels of GDF15 (Melvin et al, 2019), although the role of the GDF15-GFRAL axis in ME/CFS remains to be elucidated. In accordance with our previous work (Ost et al, 2020), we here demonstrate an involvement of the GDF15-GFRAL axis in the induction of metabolic flexibility, which is likely because of the induction of daytime-restricted anorexia, similar to the beneficial metabolic health effects of time-restricted feeding in preclinical models and humans (Regmi & Heilbronn, 2020).

With this work we further demonstrate that chronic activation of the GFRAL receptor in TG mice leads to an induction of hypothalamic *Crh*, without further activation of the HPA axis. It was recently shown that an acute elevation of endogenous or recombinant GDF15 in mice and rats leads to an activation of the HPA axis evidenced by highly increased corticosterone levels after treatment (Cimino et al, 2021), although the specific role of GFRAL in eliciting these effects remained to be elucidated. Although our data clearly support the notion of an induction of hypothalamic CRH by GDF15-GFRAL signaling, our mouse model of chronic elevated GDF15 shows that activation of the GDF15-GFRAL pathway by mitochondrial stress does not induce a sustained increase of corticosterone levels, highlighting the differential effects of acute versus chronic activation of the GDF15-GFRAL pathway. Moreover, to be best of our knowledge, the impact of acute or chronic increased circulating GDF15 levels on the well-known robust circadian oscillation of glucocorticoids (Oster et al, 2017) remains unknown and to be elucidated in future studies. Apart from its endocrine role in HPA axis signaling, hypothalamic CRH has long been recognized as a catabolic mediator, suppressing food intake in animals and humans (Rothwell, 1990; Richard & Baraboi, 2004). CRH effects on appetite and satiety signaling are largely mediated by CRH receptor 1 (CRHR1) activation (Lemos et al, 2012). Interestingly, CRHR1-knockout mice have a light phase-restricted induction of food intake (Muller et al, 2000), namely, the opposite phenotype to that observed in TG mice. There is increasing evidence that CRH neurons of the PVN are central players not only in appetite regulation but also in linking stress and anxiety behavior (Daviu et al, 2019). Thus, it is tempting to hypothesize that GFRAL-induced hypothalamic CRH signaling might modulate both anxiety-related and ingestive behavior under conditions of mitochondrial stress, but this will have to be addressed in future research, presumably with the use of CRHR1- and CRH receptor 2 (CRHR2)-knockout mouse models.

Interestingly, a recent study in mice provided a link between anxiety and systemic metabolic activation, showing that increased activities in anxiogenic circuits promote a lean phenotype, obesity resistance, and white fat browning (Xie et al, 2019). Although there are only few reports available on anxiety-like behavior in patients with mitochondrial disease, they often display psychiatric conditions including major depression and generalized or social anxiety syndromes independent of disease progression (Mancuso et al, 2013). Indeed, there appears to be a high prevalence of psychiatric symptoms observed in patients with mitochondrial mutations, which has both etiologic and therapeutic relevance (Inczedy-Farkas et al, 2012). Thus, the importance of a targeted screen for psychiatric symptoms in individuals with primary mitochondrial disease was highlighted recently (Parikh et al, 2017), although it is yet unresolved whether those symptoms are specifically related to mitochondrial disease or to other factors that are common in chronic disease conditions. Here, we provide first experimental evidence for a direct link between mitochondrial dysfunction in a peripheral (non-brain) tissue with anxiety-like behavior, potentially via hypothalamic CRH signaling in a GFRAL-dependent manner. Hence, we uncover a novel role of the GDF15-GFRAL axis that may potentially link anorectic and anxiogenic behavior in response to a chronic muscle-specific mitochondrial dysfunction.

Overall, although future studies are important to validate these results in other mouse models of mitochondrial stress, our data highlight the role of mitochondrial stress-driven endocrine crosstalk via GFRAL signaling, which may potentially enable to develop tailored disease-modifying therapeutics targeting energy balance as well as psychiatric symptoms in patients with mitochondrial disease.

---

of the figure correspond to male and female mice, respectively. Data correspond to wild-type (WT), *Gfral*-KO (GfKO), HSA-*mUcp1*-TG (TG), and HSA-*mUcp1*-TGx*Gfral*-KO (TGxGfKO) mice. Data are presented as mean + SEM. *$P < 0.05$; **$P < 0.01$; ***$P < 0.001$; ****$P < 0.0001$. Statistical test: one-way ANOVA.

Although our study provides new insights into the role of the GFRAL receptor under skeletal muscle mitochondrial stress, it presents some limitations that should be acknowledged. On the one hand, TG mice induce other hormones and cytokines, such as fibroblast growth factor 21 (FGF21) (Ost et al, 2016), that could mediate potential compensatory effects on metabolic remodeling of adipose tissue. On the other hand, future studies are required to unravel the sex-specific differences and diurnal variation of energy metabolism upon mitochondrial stress as well as a potential HPA axis induction in a circadian manner. Recent data further demonstrated the importance of the tissue specificity and dose dependency of mitochondrial integrated stress response (Croon et al, 2022). Hence, in line with the variability of mitochondrial disease manifestations seen in patients (Suomalainen & Battersby, 2018), the here-identified mitochondrial stress-induced GDF15-GFRAL axis might be differently regulated in other mouse models of tissue-specific mitochondrial stress. Notably, in this study, we employed a whole-body *Gfral*-knockout mouse. Although GFRAL has been described to be exclusively expressed in the mouse hindbrain (Hsu et al, 2017; Luan et al, 2019), we cannot exclude expression in other mouse cell types yet not discovered.

Finally, our data indicate a weak involvement of GFRAL signaling in inducing anxiety-like behavior assessed by OFT and EPM in female mice. Nevertheless, female mice do present a GFRAL-dependent induction of hypothalamic CRH, which has been often been linked to increased anxiety-like behaviors (Zhang et al, 2017). Thus, it is a possibility that these tests are not adequate for assessing female behavior of TG mice and other behavioral testing such as home cage behavioral monitoring might have to be considered for future research.

# Materials and Methods

### Animals

Mice with a C57BL/6J background were used for all experiments. *Gfral* heterozygous mice were purchased from Mutant Mouse Regional Resource Centers (MMRRC) and back-crossed to a C57BL/6J background. Mice were fed a standard chow diet (Sniff) with ad libitum access. Mice were kept group-housed and random-caged until euthanasia at 20 wk of age, when organs were collected. All animal experiments were approved by the Ethics Committee of the Ministry of Agriculture and Environment (permission number 2347-16-2020).

### Behavioral testing

The OFT was performed at 10 wk, and the EPM test was performed at 12 wk of age, both for a duration of 10 min. The open field apparatus consisted of a 50 × 50 cm enclosure. The mouse was placed in the center of the field and recorded with a camera using the software ANY-maze 5.2, which was also used for analysis of the different parameters. The EPM apparatus consisted of two open (30 × 5 × 0.5 cm) and two closed (30 × 5 × 15 cm) arms, crossing each other in a middle platform (5 × 5 × 0.5 cm). To start the test, mice were placed

in one of the open arms and were recorded for using ANY-maze 5.2, which was used as well for data analysis.

### In vivo metabolic phenotyping

Body composition was measured with quantitative magnetic resonance (QMR, EchoMRI 2012 Body Composition Analyzer). The respiratory quotient ($RQ = CO_2$ produced$/O_2$ consumed) was measured by indirect calorimetry with simultaneous recording of food intake (TSE PhenoMaster, TSE Systems).

### Gene expression analysis

RNA was isolated with a phenol-chloroform–based extraction using peqGOLD Trifast (#732-3314; VWR) followed by a DNase digest (#EN0521; Thermo Fisher Scientific). Synthesis of cDNA was performed with the LunaScript RT SuperMix Kit (#E3010L; NEB). For quantitative real-time PCR (qPCR) analyses, 5 ng of cDNA, LUNA Universal Probe qPCR Mastermix (#M3004E; NEB), and 1.5 $\mu$M of primers in a total volume of 5 $\mu$l were used. Measurements were performed on a ViiA 7 Real-Time PCR System from Applied Biosystems. The following primer sequences were used: *Gfral*: 5′-CGAAATGATGAATTATGCAGGA-3′ (F), 5′-TGCAGGTCTCATCTTCATGG-3′ (R); *Ret*: 5′-GATGGAGAGGCCAGACAACTGCA-3′ (F), 5′-CTAGAATCTAGTAAATGCATG-3′ (R); *Gdf15*: 5′-GAGCTACGGGGTCGCTTC-3′ (F), 5′-GGGACCCCAATCTCACCT-3′ (R); *18S*: 5′-CTTAGAGGGACAAGTGGCGTTC-3′ (F), 5′-CGCTGAGCCAGTCAGTGTAG-3′ (R); *Atf4*: 5′-GGAATGGCCGGCTATGG-3′ (F), 5′-TCCCGGAAAAGGCATCCT-3′ (R); *Atf5*: 5′-CTACCCCTCCATTCCACTTTCC-3′ (F), 5′-TTCTTGACTGGCTTCTCACTTGTG-3′ (R); *Atf6*: 5′-CTTCCTCCAGTTGCTCCATC-3′ (F), 5′-CAACTCCTCAGGAACGTGCT-3′ (R); *Chop*: 5′-AGAGTGGTCAGTGCGCAGC-3′ (F), 5′-CTCATTCTCCTGCTCCTTCTCC-3′ (R); *Fgf21*: 5′-GCTGCTGGAGGACGGTTACA-3′ (F), 5′-CACAGGTCCCCAGGATGTTG-3′ (R); ′ (R) *Pomc*: 5′-AACCTGCTGGCTTGCATC-3′ (F), 5′-GACCCATGACGTACTTCCG-3′ (R); *Agrp*: 5′-TTGGCGGAGGTGCTAGAT-3′ (F), 5′-ACTCGTGCAGCCTTACACAG-3′ (R); *Crh*: 5′-CAACCTCAGCCGGTTCTGAT-3′ (F), 5′-CAGCGGGACTTCTGTTGAGA-3′ (R).

### Plasma analyses

Whole blood was collected through heart puncture in heparin tubes (#41.1503.005; Sarstedt), centrifuged at 9,000*g* for 10 min at 4°C, and plasma was stored at −80°C. Plasma GDF15 was quantified using the Mouse/Rat GDF-15 Quantikine ELISA Kit (#MGD150; Bio-Techne). Plasma corticosterone was measured with a Corticosterone ELISA kit (#ADI-900-097; Enzo). Plasma ghrelin, PYY, and leptin were measured with a Meso-Scale Discovery (MSD) multiplex assay (MSD Instruments).

### Immunoblotting

Quadriceps tissue was homogenized in RIPA buffer containing protease and phosphatase inhibitor cocktail (#A32959; Thermo Fisher Scientific), and total protein content was measured with the DC Protein Assay Reagent (#500-0114; Bio-Rad). The primary antibodies used were phospho-eIF2$\alpha$ (Ser51) (#3597; Cell Signaling Technology) and eIF2$\alpha$ (#3524; Cell Signaling Technology), followed by incubation with anti-rabbit IgG (#7074; Cell Signaling Technology)

as secondary antibody. Quantification of relative protein expression was performed with ImageJ.

## Statistical analysis

Statistical analyses were performed using GraphPad Prism 9 (GraphPad Software). All data are expressed as mean with SEM. Data were tested for normality using D'Agostino and Pearson normality test. A one-way ANOVA followed by the Tukey's multiple comparison test was used to determine differences between genotypes. At $P <$ 0.05 statistical difference was assumed and denoted by *$P < 0.05$, **$P < 0.01$, ***$P < 0.001$, ****$P < 0.0001$. In addition to individual data, data are shown as mean + SEM.

# Supplementary Information

# Acknowledgements

This work was supported by grant from NutriAct (Research Stimulus Grant to C Igual Gil). Further, this work was funded by the Deutsche Forschungsgemeinschaft (DFG, German Research Foundation) – 491394008. The authors would like to thank Carolin Borchert, Petra Albrecht, and Antje Sylvester for excellent technical assistance. The graphical abstract and cartoons in Figs 1 and 4 were created with BioRender.

## Author Contributions

C Igual Gil: resources, data curation, formal analysis, validation, investigation, visualization, methodology, project administration, and writing—original draft, review, and editing.
BM Coull: investigation and writing—review and editing.
W Jonas: methodology.
RN Lippert: validation, investigation, and writing—review and editing.
S Klaus: conceptualization, supervision, funding acquisition, and writing—review and editing.
M Ost: conceptualization, formal analysis, supervision, validation, investigation, visualization, project administration, and writing—original draft, review, and editing.

## Conflict of Interest Statement

The authors declare that they have no conflict of interest.

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
