## [Reviewer comments · Life Science Alliance]

Life Science Alliance

Mitochondrial stress-induced GFRAL signaling controls diurnal food intake and anxiety-like behavior

Carla Igual Gil, Bethany Coull, Wenke Jonas, Rachel Lippert, Susanne Klaus, and Mario Ost

DOI: <https://doi.org/10.26508/lsa.202201495>

Corresponding author(s): Mario Ost, German Institute of Human Nutrition in Potsdam and Susanne Klaus, German Institute of Human Nutrition in Potsdam (DIfE)

Review Timeline:

Submission Date:	2022-04-21
Editorial Decision:	2022-06-10
Revision Received:	2022-07-30
Editorial Decision:	2022-08-15
Revision Received:	2022-08-23
Accepted:	2022-08-23

Scientific Editor: Novella Guidi

Transaction Report:

June 10, 2022

Re: Life Science Alliance manuscript #LSA-2022-01495-T

Dr. Mario Ost
German Institute of Human Nutrition in Potsdam (DIfE)
Energy Metabolism
Arthur-Scheunert-Allee 114-116
Nuthetal 14558
Germany

Dear Dr. Ost,

Thank you for submitting your manuscript entitled "Mitochondrial stress-induced GFRAL signaling controls diurnal food intake and anxiety-like behavior" to Life Science Alliance. The manuscript was assessed by expert reviewers, whose comments are appended to this letter. We invite you to submit a revised manuscript addressing the Reviewer comments.

Thank you for this interesting contribution to Life Science Alliance. We are looking forward to receiving your revised manuscript.

Sincerely,

B. MANUSCRIPT ORGANIZATION AND FORMATTING:

Reviewer #1 (Comments to the Authors (Required)):

The manuscript entitled Mitochondrial stress-induced GFRAL signaling controls diurnal food intake and anxiety-like behavior submitted by Gil et al., describes the effects of the GDF15 receptor GFRAL that goes beyond the appetite-related effects previously described for this receptor. The manuscript makes use of a mouse model of mitochondrial dysfunction with chronically elevated GDF15 levels. In this manuscript, this mouse line is crossed with a GFRAL KO mouse line, creating unique possibilities to investigate the effects of GDF15-GFRAL signaling. The authors find that GFRAL signaling induces CRH expression in hypothalamus in this model, and mediates anxiety-like behavior in male mice. Further, GFRAL is responsible for the differences observed in feeding across the light/dark cycle. The manuscript is well written and the work meticulously performed. It contributes to our understanding of GDF15-GFRAL signaling. While I find the manuscript suitable for publication, I do have some minor suggestions and comments:

1. Several studies have found that plasma GDF15 levels (markedly lower than levels presented here), is affecting food intake during 24 hrs in a GFRAL-dependent fashion (e.g. Coll et al., 2020; Breen et al., 2021). But in your model of muscle-derived GDF15, there is apparently no (24 hr/chronic) GFRAL-dependent effects on food-intake from the chronically elevated GDF15 levels. Have the authors any explanations for this discrepancy?
2. The paragraph in the Discussion "Finally, the data provided for the involvement of GFRAL in inducing anxiety-like behavior in female mice are not quite conclusive, likely due to experimental inconsistencies in the replication of the tests, which were performed in different seasons of the year, a fact that could have affected replicability of the results [58], or due to female sexhormones as mentioned above [43]. should be rephrased. Why should it not be conclusive in female mice? The findings are negative, but they might be true. If you believe that some of your assays have "experimental inconsistencies due to different seasons", then that should have been tested, e.g. January versus June; light vs dark phase - males and females. Likewise, the comment on sexhormones is valid, but could have been investigated by measuring sexhormones in this mouse model.
3. On the same note - it is puzzling that male and female mice has exact same upregulation of CRH in the hypothalamus, while only male mice display an effect in the open field test. Yet, you claim a possible association between these two. Can the authors elaborate on this?
4. And further: Cimino et al, (ref.29 in your manuscript) finds that the GDF15-mediated increase in CRH induces a rise in corticosterone levels, while you do not find any changes. Is there an explanation for this discrepancy? And is that related to 1)?
5. It would be beneficial to see the data from the OFT as percentage of time - or related to total locomotor activity. Maybe it can be included as a supplementary figure?
6. For future work: It would be interesting to see the behavioral data reproduced in another anxiety paradigm, such as the elevated plus-maze, where at least some of the parameters rely less on differences in locomotor activity.
7. Line 156: ".....social nature of mitochondria....." - what does that mean?

Reviewer #2 (Comments to the Authors (Required)):

General Comments

Gil and colleagues describe a clear set of measurements to investigate the role of GDF15 in the whole-body metabolic rewiring response to tissue-specific mitochondrial stress. First, the authors cross a previously published mouse model of skeletal muscle mitochondrial stress with a GFRAL knock out mouse. Second, the authors characterize the degree of physiological and behavioral rescue provided by GFRAL KO in male and female transgenic mice. All experiments were performed in both female and male mice, which is laudable and important, and is a strength of the work. Overall, the manuscript is well-written and the figures are well-structured, providing a compelling argument for the role of GFRAL in mediating the effects of muscle-derived

GDF15 on the whole body respiratory quotient, appetite regulation, HPA axis regulation, and anxiety-like behavior. This study addresses an important knowledge gap in the literature and could be enhanced in several ways.

Specific Comments

1. The authors use 24-hour indirect calorimetry to measure metabolic flexibility in the mice. Energy expenditure quantification over the 24-hour measurement period from those data would provide insight into body weight rescue phenotype observed in the TGxGfKO mice and strengthen the manuscript. Do the TG animals show elevated energy expenditure, and is this rescued in TGxGfKO mice? The authors should show energy expenditure, ideally with the same level of resolution as the RQ (Figure 2a,c), across the 24-hour period. Ideally, energy expenditure should be normalized to fat-free mass if available, and data normalized to total body weight would also be useful if the total body weight of the animals is different. Group differences should be analyzed separately for the light and dark phases of the light cycle, as in Figure 2f and h. These analyses would deserve an additional supplementary figure.
2. The results in Figure EV2 appear to overlap with the control and TG results from Figure 3, which shows the same result but with the inclusion of TGxGfKO mice. Can the authors clarify the difference between the two sets of experiments?
3. Why were mice caged differently prior to measurement in Figure EV2? Could this be a contributing factor to the differences in the Open-field-test results in female WT and TG animals between Figure 3h and Figure EV2h?
4. Can the authors provide more description of the significance of metabolic flexibility and interpretation of their result in the discussion?
5. Statistics: Repeated-measures ANOVA is not the correct test to apply to these groups of animals as there are different individuals within each group. One way ANOVA (or a mixed-effects model if the data do not meet the criteria for distribution normality) would be a more appropriate test.
6. The original publications on GFRAL paints a fairly superficial but compelling picture that it is primarily or exclusively expressed in the mouse brainstem. However, the Allen Mouse Brain dataset also shows non-zero expression in other brain regions (<https://www.proteinatlas.org/ENSG00000187871-GFRAL/summary/rna>). The pig brain atlas also shows expression across at least a dozen cortical and subcortical regions with no preferential expression in the brainstem (same link as above). Expression data in humans suggest also indicates that GFRAL is expressed in most brain regions. It may also be expressed in immune cells, at least at the RNA level. What is the rationale for using a whole-body GFRAL KO model, rather than a brain-specific or even brainstem-specific system? If the authors are convinced of the brainstem specificity of GFRAL, at least in mice, they should provide a compelling argument and rationale for using a whole body KO. If they are not entirely convinced, they should discuss the more widespread GFRAL expression across tissues beyond the brainstem and point out the implications of this limitation in relation to their interpretation. This caveat does not invalidate any of the main findings, but overseeing this could propagate an overly limited notion regarding the site of action for GFRAL. For example, could GDF15 be having some of its effects via signaling on immune cells, which in turn influence brain and behavioral processes? Testing this possibility by generating immune cell GFRAL-KO animals would be a strong way to rule out this possibility, although this is likely beyond the scope of this study.
7. The authors rely on the results of their 2020 EMBO publication to validate their muscle specific mitochondrial stress model. The induction of GDF15 is the same with or without GFRAL, which is an important control. Have the authors verified that the muscle mitochondrial phenotype is the same as in their previous work?

Minor Comments

8. The figures at the end of the manuscript are not labeled/numbered, which makes the reviewer's job difficult. Best would be to have the figures with their legends embedded in the text.
9. Lines 163-164: Sentence is unclear regarding effect on lifespan.

Reviewer comments

Reviewer #1:

The manuscript entitled Mitochondrial stress-induced GFRAL signaling controls diurnal food intake and anxiety-like behavior submitted by Gil et al., describes the effects of the GDF15 receptor GFRAL that goes beyond the appetite-related effects previously described for this receptor. The manuscript makes use of a mouse model of mitochondrial dysfunction with chronically elevated GDF15 levels. In this manuscript, this mouse line is crossed with a GFRAL KO mouse line, creating unique possibilities to investigate the effects of GDF15-GFRAL signaling. The authors find that GFRAL signaling induces CRH expression in hypothalamus in this model, and mediates anxiety-like behavior in male mice. Further, GFRAL is responsible for the differences observed in feeding across the light/dark cycle. The manuscript is well written and the work meticulously performed. It contributes to our understanding of GDF15-GFRAL signaling. While I find the manuscript suitable for publication, I do have some minor suggestions and comments:

Response: Thank you for your interest in our research. We have worked to address all of your comments and remarks by improving our discussion as well as including additional data. We believe that these changes have strengthened our manuscript and hope that you find it suitable for publication.

1. Several studies have found that plasma GDF15 levels (markedly lower than levels presented here), is affecting food intake during 24 hrs in a GFRAL-dependent fashion (e.g. Coll et al., 2020; Breen et al., 2021). But in your model of muscle-derived GDF15, there is apparently no (24 hr/chronic) GFRAL-dependent effects on food-intake from the chronically elevated GDF15 levels. Have the authors any explanations for this discrepancy?

Response: Point taken well. Based on the results of our current and previous study (PMID: 32026535), we cannot explain why muscle-derived GDF15 is promoting a daytime-restricted food intake upon chronic mitochondrial stress. We have previously hypothesized that the anorectic action requires a certain threshold of circulating GDF15 (≥ 400 pg/ml). In line with that, metformin treatment of chow fed lean mice only promotes a mild increase in circulating GDF15 levels (~ 150 - 200 pg/ml) and no food intake reduction (PMID: 31875646 Extended Data Figure 2 + Klein et al. *bioRxiv* 2022 Figure 1d; <https://doi.org/10.1101/2022.02.16.480373>). In obese mice, concentration of GDF15 reaches ~ 1000 pg/mL when treated with daily oral doses of metformin, which indeed coincides with body weight reduction (PMID: 31875646, Klein et al. *bioRxiv* 2022). However, while metformin does reduce food intake in those obese mice, the mechanistical role of the GDF15-GFRAL axis remains controversial as Klein et al. demonstrated that metformin lowers body weight independent of the GDF15-GFRAL pathway. Furthermore, two independent studies recently demonstrated that induced GDF15 is neither required for the LPS-induced weight loss and anorexia (PMID: 34189431) nor an LPS-mediated corticosterone induction in mice (PMID: 34187898). Hence, although circulating GDF15 levels are commonly increased under different conditions (e.g. cellular & mitochondrial stress; treatment with LPS, metformin or cisplatin; exercise; aging; obesity; cancer), the specific impact on food intake regulation appears less consistent and remains puzzling. Therefore, we do believe that our data presented here are not at discrepancy with the literature but rather expand our view on the variety of a chronic GDF15 action via its receptor GFRAL by inducing a shift in the diurnal pattern of food intake.

2. The paragraph in the Discussion "Finally, the data provided for the involvement of GFRAL in inducing anxiety-like behavior in female mice are not quite conclusive, likely due to experimental inconsistencies in the replication of the tests, which were performed in different seasons of the year, a fact that could have affected replicability of the results [58], or due to female sex hormones as mentioned above [43]. should be rephrased. Why should it not be conclusive in female mice? The findings are negative, but they might be true. If you believe that some of your assays have "experimental inconsistencies due to different seasons", then that should have been tested, e.g. January versus June; light vs dark phase - males and females. Likewise, the comment on sex hormones is valid, but could have been investigated by measuring sex hormones in this mouse model.

Response: Thank you for the excellent comment. We acknowledge the importance of testing the effect of the season in the female behavior, but due to our animal protocol and experimental procedure this was beyond our possibilities. However, we have added additional data from EPM testing (see revised Figure 4 and Figure EV4) that confirm our negative results from OFT, suggesting

Rebuttal letter #LSA-2022-01495-T

that female TG mice only show a very mild anxiety-like behavior compared to male TG littermates. Therefore, we have rephrased that paragraph accordingly (see lines 269-274).

3. On the same note - it is puzzling that male and female mice has exact same upregulation of CRH in the hypothalamus, while only male mice display an effect in the open field test. Yet, you claim a possible association between these two. Can the authors elaborate on this?

Response: Thank you for pointing this out. Indeed, our data suggest that females perform better within the experimental test environment (OFT, EPM), as described by others (PMID: 11240006), while under basal physiological conditions (e.g. timepoint of dissection) both males and females show a CRH induction. Genetic ablation of hypothalamic CRH expression has been previously shown to reduce anxiety-like behavior in mice (PMID: 27595593). Whether a chronic CRH induction is relevant for a metabolic and/or behavioral response under muscle mitochondrial stress conditions in females remains to be elucidated. Although out of the scope of our study, this could be achieved by further behavioral testing within a home-cage setting (e.g. using the TSE IntelliCage for 24/7 behavioral and cognitive phenotyping). We have now rephrased our discussion addressing the connection of CRH induction with female mice behavior and important steps in future research in order to clarify this issue (see lines 269-274).

4. And further: Cimino et al, (ref.29 in your manuscript) finds that the GDF15-mediated increase in CRH induces a rise in corticosterone levels, while you do not find any changes. Is there an explanation for this discrepancy? And is that related to 1)?

Response: Thanks for raising this important question. We believe that this discrepancy arises from the fact that we are dealing with a pathophysiological, chronic GDF15-GFRAL pathway activation, which strongly differs from the models used by Cimino and colleagues. In their work, the authors use recombinant GDF15 as well as the GDF15-inducing compounds LPS, tunicamycin and cisplatin and indeed, opposite to our findings, show increases in plasma corticosterone after 1-6h treatment. We believe that, when the GDF15-GFRAL axis is induced chronically, there is an adaptation to this signal that prevents constantly high corticosterone levels. Nevertheless, and since corticosterone is known to be tightly regulated by circadian cues (PMID: 27749086), it is a possibility that under these chronic conditions, corticosterone levels are higher in TG mice at other timepoints of the 24 hr cycle. Future studies should, therefore, address circadian changes in corticosterone levels in TG mice. We have now included a sentence in our discussion in order to further clarify the discrepancy between the work by Cimino et al and our study (lines 219-222).

5. It would be beneficial to see the data from the OFT as percentage of time - or related to total locomotor activity. Maybe it can be included as a supplementary figure?

Response: Thank you for pointing this out. We agree that locomotor activity is an essential parameter of an OFT. We have now included the total distance travelled during the OFT (see revised Figure EV4). Since total distance was not changed amongst the genotypes, normalization of the other parameters did not alter the results and therefore we did not include these data in the manuscript. Regarding expressing the data as a percentage of time, we agree that this is a better way to present it. We now exchanged absolute time by percentage of time in all OFT data as well as the newly included elevated plus maze (EPM) data (see revised Figure 4 and Figure EV4).

6. For future work: It would be interesting to see the behavioral data reproduced in another anxiety paradigm, such as the elevated plus-maze, where at least some of the parameters rely less on differences in locomotor activity.

Response: Thanks for this remark. We performed an elevated plus maze (EPM) test and have now included the data in the manuscript (see revised Figure 4), which we believe strengthen the conclusions made from the OFT. We have as well included plasma corticosterone measurements after the EPM that, in our opinion, further supports the sex-specific differences in behavioral cues (see revised Figure EV4).

7. Line 156: ".....social nature of mitochondria....." - what does that mean?

Response: Sorry for the confusion. This sentence has been rephrased (line 178)

Reviewer #2:
General Comments

Gil and colleagues describe a clear set of measurements to investigate the role of GDF15 in the whole-body metabolic rewiring response to tissue-specific mitochondrial stress. First, the authors cross a previously published mouse model of skeletal muscle mitochondrial stress with a GFRAL knock out mouse. Second, the authors characterize the degree of physiological and behavioral rescue provided by GFRAL KO in male and female transgenic mice. All experiments were performed in both female and male mice, which is laudable and important, and is a strength of the work. Overall, the manuscript is well-written and the figures are well-structured, providing a compelling argument for the role of GFRAL in mediating the effects of muscle-derived GDF15 on the whole-body respiratory quotient, appetite regulation, HPA axis regulation, and anxiety-like behavior. This study addresses an important knowledge gap in the literature and could be enhanced in several ways.

Response: Thank you for your positive response to our work We have modified the manuscript according to your suggestions and remarks and believe it is considerably improved. The specific changes are addressed below.

Specific Comments

1. The authors use 24-hour indirect calorimetry to measure metabolic flexibility in the mice. Energy expenditure quantification over the 24-hour measurement period from those data would provide insight into body weight rescue phenotype observed in the TGxGfKO mice and strengthen the manuscript. Do the TG animals show elevated energy expenditure, and is this rescued in TGxGfKO mice? The authors should show energy expenditure, ideally with the same level of resolution as the RQ (Figure 2a,c), across the 24-hour period. Ideally, energy expenditure should be normalized to fat-free mass if available, and data normalized to total body weight would also be useful if the total body weight of the animals is different. Group differences should be analyzed separately for the light and dark phases of the light cycle, As in Figure 2f and h. These analyses would deserve an additional supplementary figure.

Response: We appreciate your comment. As suggested, we have included an additional supplementary figure (EV2) addressing the role of GFRAL on the energy expenditure phenotype of TG mice.

2. The results in Figure EV2 appear to overlap with the control and TG results from Figure 3, which shows the same result but with the inclusion of TGxGfKO mice. Can the authors clarify the difference between the two sets of experiments?

Response: Thank you for the remark. The former figure EV2 that you refer to here corresponded to a pilot study that we performed in order to screen for any anxiety-like phenotype in TG mice. We agree though, that data presented in former Figures EV2 and Figure 3 were overlapping and have decided to omit data from the pilot study (former Figure EV2). Instead, and attending as well to point 6 of Reviewer #1, we have extended the phenotypical characterization in the revised Figure 4 by adding data from an elevated plus maze test.

3. Why were mice caged differently prior to measurement in Figure EV2? Could this be a contributing factor to the differences in the Open-field-test results in female WT and TG animals between Figure 3h and Figure EV2h?

Response: We apologize for the misunderstanding. Mice presented in former figure EV2 were just single-caged one week prior to their sacrifice at 20 weeks. Nevertheless, the open field test was performed at 10 weeks of age, and at that timepoint all mice were group-housed prior to the test and in the exact same conditions than the cohort presented in former Figure 3. However, as pointed out in our response to point 2, we have now decided to remove the data from former Figure EV2 in order to avoid confusion.

4. Can the authors provide more description of the significance of metabolic flexibility and interpretation of their result in the discussion?

Response: Thank you for your comment. We have now included an interpretation of this result in the discussion (lines 207-210).

5. Statistics: Repeated-measures ANOVA is not the correct test to apply to these groups of animals as there are different individuals within each group. One-way ANOVA (or a mixed-effects model if the data do not meet the criteria for distribution normality) would be a more appropriate test.

Response: We apologize for this mistake. The test performed was indeed a one-way ANOVA. We corrected this in the manuscript.

6. The original publications on GFRAL paints a fairly superficial but compelling picture that it is primarily or exclusively expressed in the mouse brainstem. However, the Allen Mouse Brain dataset also shows non-zero expression in other brain regions (<https://www.proteinatlas.org/ENSG00000187871-GFRAL/summary/rna>). The pig brain atlas also shows expression across at least a dozen cortical and subcortical regions with no preferential expression in the brainstem (same link as above). Expression data in humans suggest also indicates that GFRAL is expressed in most brain regions. It may also be expressed in immune cells, at least at the RNA level. What is the rationale for using a whole-body GFRAL KO model, rather than a brain-specific or even brainstem-specific system? If the authors are convinced of the brainstem specificity of GFRAL, at least in mice, they should provide a compelling argument and rationale for using a whole-body KO. If they are not entirely convinced, they should discuss the more widespread GFRAL expression across tissues beyond the brainstem and point out the implications of this limitation in relation to their interpretation. This caveat does not invalidate any of the main findings, but overseeing this could propagate an overly limited notion regarding the site of action for GFRAL. For example, could GDF15 be having some of its effects via signaling on immune cells, which in turn influence brain and behavioral processes? Testing this possibility by generating immune cell GFRAL-KO animals would be a strong way to rule out this possibility, although this is likely beyond the scope of this study.

Response: We highly appreciate this point. We have now included a multi-tissue expression analysis for *Gfral* mRNA expression (see Figure EV1), showing that *Gfral* is only expressed in the hindbrain (AP/NTS), as previously described (PMID: 28846099; PMID: 28846098; PMID: 31402172). Importantly, expression was not induced in any tissue surveyed by transgenic UCP1 expression in the muscle providing a rationale for using a whole-body *Gfral*-KO. Hence, although we cannot rule out a possible GFRAL expression in certain specific cell types, any effect of manipulating the muscle-derived GDF15 signaling pathway in TG mice is likely mediated through activation of GFRAL-expressing neurons in the area postrema. We have now included this rationale in the discussion (lines 266-268).

7. The authors rely on the results of their 2020 EMBO publication to validate their muscle specific mitochondrial stress model. The induction of GDF15 is the same with or without GFRAL, which is an important control. Have the authors verified that the muscle mitochondrial phenotype is the same as in their previous work?

Response: Thank you for raising this point. We have now included one more figure in our manuscript (see revised Figure 2) addressing the involvement of GFRAL signaling in the muscle mitochondrial phenotype of TG mice.

Minor Comments

8. The figures at the end of the manuscript are not labeled/numbered, which makes the reviewer's job difficult. Best would be to have the figures with their legends embedded in the text.

Response: Thank you for this remark. We have now included the figures above their corresponding figure legends in the submitted revised manuscript and hope it makes the review process easier.

9. Lines 163-164: Sentence is unclear regarding effect on lifespan.

Response: Thank you for this remark. We have slightly modified the sentence in the revised manuscript (lines 185-186) in order to make the point clear.

August 15, 2022

RE: Life Science Alliance Manuscript #LSA-2022-01495-TR

Dr. Mario Ost
German Institute of Human Nutrition in Potsdam
Energy Metabolism
Arthur-Scheunert-Allee 114-116
Nuthetal 14558
Germany

Dear Dr. Ost,

Thank you for submitting your revised manuscript entitled "Mitochondrial stress-induced GFRAL signaling controls diurnal food intake and anxiety-like behavior". We would be happy to publish your paper in Life Science Alliance pending final revisions necessary to meet our formatting guidelines.

- please make sure the authors order in your manuscript and our system match
- please consult our manuscript preparation guidelines <https://www.life-science-alliance.org/manuscript-prep> and make sure your manuscript sections are in the correct order
- please add ORCID ID for corresponding (and secondary corresponding) author--you should have received instructions on how to do so
- please add a Category for your manuscript in our system
- please add a Summary Blurb/Alternate Abstract in our system
- please add a conflict of interest statement to your main manuscript text
- please upload your main and supplementary figures as single files; - Please upload all figure files as individual ones, including the supplementary figure files; all figure legends should only appear in the main manuscript file
- LSA allows supplementary figures, but not EV Figures; please update your callouts for the Supplementary Figures in the manuscript Fig EV1A = Fig S1A)
- please upload your main manuscript text as an editable doc file;
- please add the Twitter handle of your host institute/organization as well as your own or/and one of the authors in our system

A. FINAL FILES:

-- Summary blurb (enter in submission system): A short text summarizing in a single sentence the study (max. 200 characters including spaces). This text is used in conjunction with the titles of papers, hence should be informative and complementary to the title. It should describe the context and significance of the findings for a general readership; it should be written in the

present tense and refer to the work in the third person. Author names should not be mentioned.

B. MANUSCRIPT ORGANIZATION AND FORMATTING:

Sincerely,

Reviewer #1 (Comments to the Authors (Required)):

The authors have sufficiently addressed all my concerns.

Reviewer #2 (Comments to the Authors (Required)):

Excellent revision. The authors have satisfactorily addressed all comments. The revised manuscript is significantly strengthened.

August 23, 2022

RE: Life Science Alliance Manuscript #LSA-2022-01495-TRR

Dr. Mario Ost
German Institute of Human Nutrition in Potsdam
Energy Metabolism
Arthur-Scheunert-Allee 114-116
Nuthetal 14558
Germany

Dear Dr. Ost,

Thank you for submitting your Research Article entitled "Mitochondrial stress-induced GFRAL signaling controls diurnal food intake and anxiety-like behavior". It is a pleasure to let you know that your manuscript is now accepted for publication in Life Science Alliance. Congratulations on this interesting work.

DISTRIBUTION OF MATERIALS:

Again, congratulations on a very nice paper. I hope you found the review process to be constructive and are pleased with how the manuscript was handled editorially. We look forward to future exciting submissions from your lab.

Sincerely,
